# Designing Automated Deployment Strategies of Face Recognition Solutions in Heterogeneous IoT Platforms

**Unai Elordi** [1,2,*], **Chiara Lunerti** [1], **Luis Unzueta** [1], **Jon Goenetxea** [1], **Nerea Aranjuelo** [1,2], **Alvaro Bertelsen** [1] **and Ignacio Arganda-Carreras** [2,3,4]

1   Vicomtech, Basque Research and Technology Alliance (BRTA), 20009 Donostia, Spain; clunerti@vicomtech.org (C.L.); lunzueta@vicomtech.org (L.U.); jgoenetxea@vicomtech.org (J.G.); naranjuelog@vicomtech.org (N.A.); abertelsen@vicomtech.org (A.B.)
2   Computer Science and Artificial Intelligence Department, Campus Donostia, University of the Basque Country (UPV/EHU), 20018 Donostia, Spain; ignacio.arganda@ehu.eus
3   Ikerbasque, Basque Foundation for Science, 48009 Bilbao, Spain
4   Donostia International Physics Center (DIPC), 20018 Donostia, Spain
*   Correspondence: uelordi@vicomtech.org

**Abstract:** In this paper, we tackle the problem of deploying face recognition (FR) solutions in heterogeneous Internet of Things (IoT) platforms. The main challenges are the optimal deployment of deep neural networks (DNNs) in the high variety of IoT devices (e.g., robots, tablets, smartphones, etc.), the secure management of biometric data while respecting the users' privacy, and the design of appropriate user interaction with facial verification mechanisms for all kinds of users. We analyze different approaches to solving all these challenges and propose a knowledge-driven methodology for the automated deployment of DNN-based FR solutions in IoT devices, with the secure management of biometric data, and real-time feedback for improved interaction. We provide some practical examples and experimental results with state-of-the-art DNNs for FR in Intel's and NVIDIA's hardware platforms as IoT devices.

**Keywords:** knowledge-driven approach; face recognition; deep neural networks; Internet of Things; user interaction

## 1. Introduction

Industry 4.0, the fourth Industrial Revolution, is focused on interconnectivity, automation, autonomy, machine learning, and real-time data, in parallel to the growing number of interconnected devices around the world [1]. The interconnected devices are very diverse, and include wearables, smart appliances, smartphones, tablets, smart TVs, embedded computers, gateways, laptops, computers, different kinds of robots, etc. In this context, the Internet of Things (IoT) paradigm plays a major role, using applications running on devices with sensing, networking, and processing capabilities that interact with other devices and services on the Internet. IoT allows for the integration of the physical world into computer-based systems, providing manufacturing companies with many growth opportunities. Thus, expanding IoT capabilities will assist with the development of more sophisticated products and services to contribute to the progress of Industry 4.0.

One way to expand these products is to improve the sensing capabilities of interconnected IoT devices to detect and recognize users, so as to allow them to interact securely with IoT applications. In recent years, Deep Neural Networks (DNNs) have allowed the development of robust solutions for computer vision tasks such as face recognition (FR), including face detection, alignment, and identity recognition [2–4]. Current FR solutions can help to secure the use of IoT applications, by restricting access to sensitive data to individuals with the required permissions, in a pervasive and user-friendly way, without the need to memorize something (e.g., PINs or passwords). Users could perform authentication

by having their face recognized on IoT devices with cameras, and this one-time login could provide access to a full IoT network of devices of many different kinds. Furthermore, some applications might require these machines to identify people at a distance without their collaboration (e.g., to localize users, or for surveillance purposes), and FR opens up this possibility, unlike other biometric-based authentication alternatives, such as fingerprints, iris, or hand geometry [5].

However, deploying DNN-based FR solutions in IoT platforms is a challenging issue, mainly due to the following factors:

- To obtain (near) real-time responses, DNN models need to be processed locally, and not on remote servers, as server-device data transference would add considerable delays in such cases, the computational cost of DNN inference could be higher than the computational resources available in many IoT devices. Besides, they could have different kinds of processors (XPUs: CPUs, GPUs, FPGAs, etc.), which require specific DNN inference engines (Intel's OpenVINO, Google's TensorFlow Lite, NVIDIA's TensorRT, Facebook's PyTorch, etc.) [6].
- To allow users to enroll on one device and authenticate on another, respecting their privacy in compliance with the law, such as the EU's General Data Protection Regulation (GDPR), biometric data needs to be managed securely, preventing intruders from gaining access.
- Besides the high heterogeneity of IoT devices, with which users might interact, in terms of shape, functionalities, sensing, and computing capabilities, we might face a high variety of user-interaction capabilities, from fully active to fully assisted. All users should be able to interact satisfactorily with the deployed FR system during the face enrollment and verification stages.

In [7], we manage these three challenges for the specific case of IoT platforms for elderly care applications. In the study—following user-centered design principles [8]—we proposed an approach that relied on a series of pretrained lightweight DNN models deployed locally to perform the following tasks: Face and facial landmark detection, pose and gesture recognition, spoofing attacks detection, and identity recognition. The first two DNNs guide the user in real-time during the enrollment and verification stages and for liveness detection. An automated procedure selects the appropriate DNN inference engine, DNN model configurations, and batch size for each IoT device. To secure the biometric data, fully homomorphic encryption is used.

In this paper, we extend our approach beyond elderly care applications, by including a knowledge-driven decision-making procedure [9–11] that allows for an improvement of the automatization process of the DNN-based FR system's deployment on the IoT devices. We also reviewed the interaction and recognition workflow to improve they way feedback is delivered to the user to extract the biometric data from the facial image in the best possible conditions. We also include further experimental results with different kinds of hardware platforms—Intel's and NVIDIA's processors—and inference engines—Intel's OpenVINO and NVIDIA's TensorRT—for DNN inference.

The paper is organized as follows. Section 2 reviews the related work. Section 3 explains the proposed approach. Section 4 presents experimental results and an evaluation of the approach. Finally, Section 5 contains conclusions and ideas for future research.

## 2. Related Work

### 2.1. User Interaction on Face Recognition Systems

Blanco-Gonzalo et al. state, in [12], that security systems need to be reliable and easy to use for an as wide as possible cross-section of the population, to ensure that personal data is secure. Their purpose is to analyze whether people with accessibility concerns can perform certain tasks more easily as a result of biometric systems. To this end, they evaluated the accessibility of different user authentication mechanisms—including face verification—in a mobile app. Their experiments reveal that almost all groups of participants had difficulties in both handling the device and framing the face for this task, especially those in the

"hands/arms disabilities" group, which led to low-quality samples. This might be because test subjects took "selfies" without any face detection or image quality feedback, even though they were instructed to locate the face frontal and within the boundaries of a guiding bounding box. Nevertheless, as face recognition required less interaction than other modalities, users considered it fashionable.

In [13], Hofbauer et al. analyzed several commercial off-the-shelf face recognition systems on smartphones for device unlocking, focusing on the tradeoff between acceptance and security, which are both related to interaction. To evaluate acceptance, they used the time to unlock and vary the illumination conditions. Moreover, to evaluate the security, they used different spoofing attacks. They concluded that creating a generic computer vision-based system, without dedicated hardware—such as Apple's FaceID that uses additional sensors—is challenging. In their experiments, the main problems pertained to liveness detection, as the biometric comparison worked well in almost all cases. The main issue with liveness detection seemed to be harsh lighting conditions, which normally do not occur in lab environments used for testing during the development stage.

### 2.2. Deployment of Face Recognition Systems in IoT Platforms

The challenge of creating a secure and efficient user authentication on IoT platforms is becoming an increasingly relevant issue. The recently published works reveal the increasing interest in this topic. The comprehensive review of Yousefpour et al. [14] is usually referenced as an article that covers secure user authentication along with many other IoT-related paradigms such as fog computing, cloudlets, and multi-access edge computing (MEC). All these approaches handle the security and the performance of the data produced by the IoT platform.

Hu et al. [15] present an example of a fog-based face identification and resolution system. Instead of using DNNs for images processing, this work proposes classic Haar and Local Binary Pattern (LBP) features for face detection and identification. Although these methods use less computational resources, these computer vision methods are not as accurate as DNN face recognition techniques. To reduce the network traffic, this framework delegates the part of the resolution task to fog nodes, and only the biometric data is transmitted to the cloud. Using a task partitioning strategy, the cloud overhead is relieved and the devices located at the edge network assume the role of image processing, making full use of the computing power. In a previous work [16], Hu et al. extended this approach to solve confidentiality, integrity, and availability issues. The proposed method provides a mechanism for an authentication and session key agreement scheme, supported by a data encryption scheme and data integrity checking.

Wang and Nakachi [17] present a secured framework for face recognition in edge and cloud networks based on sparse representation. In contrast to DNN based face recognition, this method is based on a discriminative dictionary, which requires fewer computation resources, but the accuracy results are worse than DNN based face-recognition approaches. Another inconvenience of this method is that each time a new sample is added, dictionary training is required. They establish a distributed learning framework to train the recognition method. The training samples are divided into two parts, one for dictionary and classifier training, and one for ensemble training. The decision template is extracted for each class in the intermediate space, expanded by the estimated label vectors and based on the ensemble training set. The recognition is identified according to pairwise similarity between the decision profile of the testing sample and each of the decision templates. To guarantee privacy, they adopt a low complexity for the encrypting algorithm, based on random unitary transform, without affecting the accuracy.

Mao et al. [18] designed an edge device-based DNN training scheme for face recognition with differentially private mechanisms to protect private data. The DNN is split into two parts, one deployed on the user's device and the other on the edge server. They avoided cryptographic tools to keep the user side lightweight.

## 3. Proposed Approach

### 3.1. User Interaction Workflow for Face Verification

The user interaction and recognition workflow are presented in Figure 1. The user triggers this workflow the first time to enroll into the system, and for each following instance, the user's identity needs to be verified. This workflow is designed considering the user-centered design principles described in [8], and the importance of providing real-time feedback, as suggested in [12], not only to obtain good quality images but also to improve the technology acceptance and adoption.

The workflow is divided into three phases: (1) facial image acquisition, (2) spoof detection, and (3) biometric features extraction. The goal of phase 1 is to assess the user on how to present themselves to the camera to extract facial images with sufficient quality for the subsequent phases. Then, in phase 2, the system will check that there is not an attempt of spoofing, and if there is, it would ask for a new image, returning to phase 1. Finally, in phase 3, it will perform biometric feature extraction for registration or the verification. Some users could have physical difficulties that could interfere with the interaction with the device, making it difficult to hold it still when taking a picture. For this reason, the proposed interaction does not present a button to press when taking the image, as this action could cause extra movement and result in a blurred image. Instead, the system will automatically take the image when all the required quality conditions are met.

More specifically, in phase 1, the system's interface will show the mirrored video stream captured by the device's camera with an overlapping human shape that will work as a guideline for users. First, based on the face regions, facial landmarks, head pose, and facial gesture estimations obtained by a series of DNNs [19,20], the system indicates if the face is too close or far from the camera ("move closer", "move further"), or if the head is tilted ("move to the left" or "move to the right"). Since some users might have difficulties in reading small text, an icon would indicate the movement to facilitate the comprehension of the feedback messages. In the case that the user is accompanied by someone else (e.g., an assistant), the system will detect more than one face, but it will consider only the face closer to the device as the user that requires verification, hence providing the feedback to that specific user. Then, once the image passes this check, the user's facial image is cropped and "normalized" based on the detected facial landmarks (i.e., it is resized and rotated so that the eyes are in the predefined positions in the cropped image). This normalized facial image is then processed by another DNN for image quality assessment [21]. This DNN provides a score that can also be obtained to check whether the image respects the required conditions, providing feedback to improve them if they are not met (with a message such as "improve the lighting conditions"). If this check is also passed, the processing of the user's normalized facial image continues in phase 2. During phases 2 and 3 there is no need to show the user the mirror image from the camera as it could be confusing (seeing the image the whole time might lead them to think that there is something wrong with the image acquisition when it has been acquired already).

In phase 2, following the observations made in [13], to improve the security of the verification system, another DNN would verify whether the normalized facial image corresponds to a spoofing attack or not [22,23]. This procedure could be enhanced using depth sensors instead of only a camera.

In phase 3, another DNN would extract the biometric feature vector from the normalized facial image that allows for the authentication of facial identities [24]. Once the features are extracted, the system can proceed to compare them against those stored during registration with a verification algorithm—that could involve Euclidean distance, cosine similarity, etc.—and either grant or deny access, depending on the result.

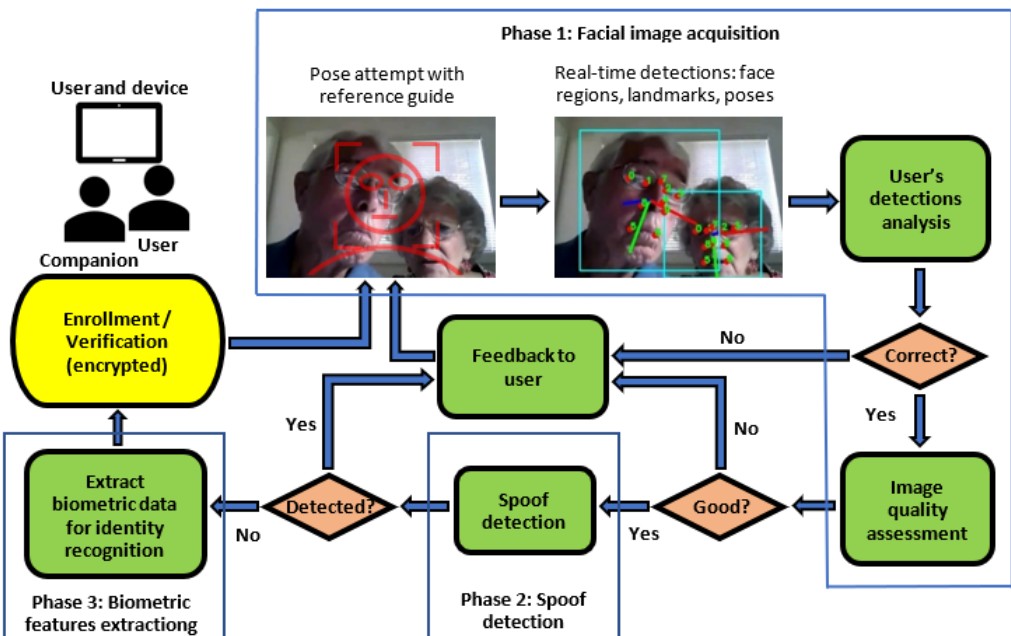

**Figure 1.** User interaction and recognition workflow.

### 3.2. Deployment of Algorithms

Figure 2 shows our proposed knowledge-driven workflow for the automated deployment of FR solutions on heterogeneous IoT platforms for industrial applications (e.g., on the private network of a company's facilities). This deployment scenario considers a wide range of devices and robots with whom users might interact.

To reduce the latency of the FR solution as much as possible, this deployment workflow considers two runtime scenarios. The first scenario contemplates on-device DNN inference for devices with DNN inference capabilities. The second scenario deploys the FR solution on the IoT gateway to manage multiple requests at once using containerized services. This deployment process categorizes the client devices and robots into suitable and unsuitable devices for DNN inference. Suitable devices will host the FR service directly, and the rest will communicate with the IoT gateway when required, by submitting images and processing requests to the gateway and receiving the corresponding responses.

To improve the user experience, users register their face credentials in one device and the biometric data is shared among the rest of the private network's IoT devices. Thus, the face login can be performed on any of these devices. Even though this network is private, biometric data encryption is necessary, so the administrator manages the encryption keys to preserve privacy, as explained in the next section.

This deployment procedure automatically decides which is the most suitable DNN package—DNN models and Inference Engine (IE)—for optimal DNN inference in each IoT device. The decisions are made by a Case-Based Reasoning (CBR) system [25]. A CBR is a problem-solving method for which the solutions are based on previous experiences. A CBR system is organized in cases, which are represented as problems and solutions. An "experience" instance would correspond to a previously solved case. In our context, we have one case type that would be represented in the following way:

- Problem: New device with heterogeneous hardware (i.e., might have one or more kinds of processors, including in some cases DNN accelerators).
- Solution: The most optimal DNN IE and DNN model configuration package for the target device.

As is shown in Figure 2, the automated deployment workflow has eight steps, and the CBR process is its core. The standard CBR process is comprised of four tasks: (1) Retrieve case, (2) reuse/adapt case, (3) evaluate case, and (4) retain case.

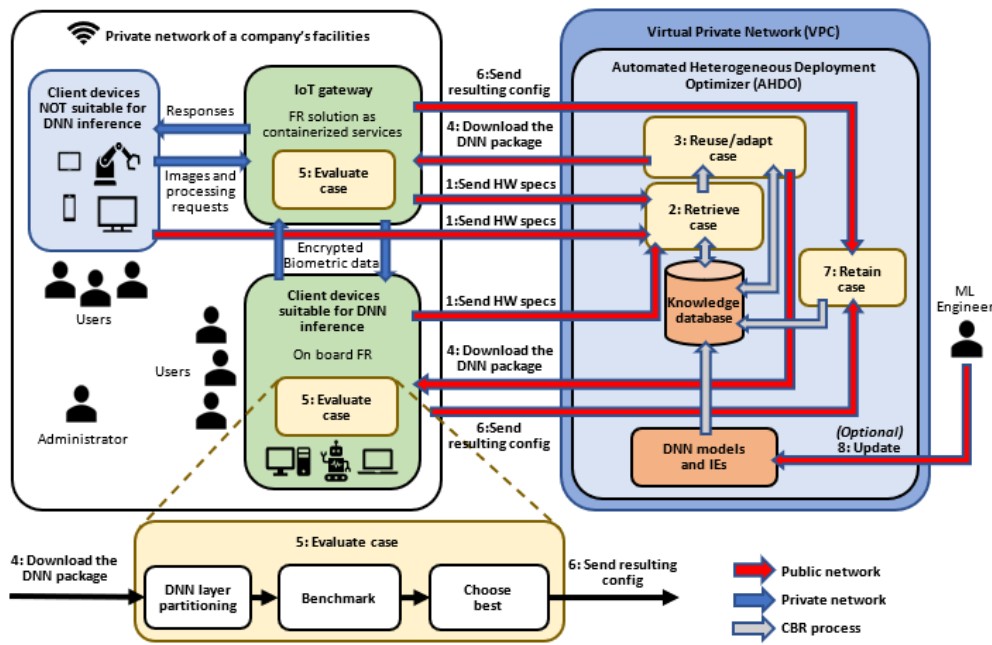

**Figure 2.** Workflow for the automated deployment of the IoT face recognition solution.

Almost all of the tasks are executed in a module called Automated Heterogeneous Deployment Optimizer (AHDO), which is hosted in a Virtual Private Cloud (VPC). In contrast, each IoT device and gateway with DNN inference capabilities execute the case evaluation task. Next, we explain all these steps and components in detail.

### 3.2.1. Case Retrieval

This task is the CBR's starting point. Its objective is to retrieve those old cases (experiences) that are most similar to the new case. Each client and the IoT gateway activate this retrieving task by sending their hardware specifications to the AHDO (step 1). Each hardware specification represents a new case for the CBR system. The old cases are stored in the knowledge database. All cases (new, old) follow a specific case structure which we define in categorical and quantitative values.

The categorical values represent the hardware vendor (Intel, NVIDIA, Google), the DNN accelerator types (GPU, CPU, VPU, TPU, FPGA, etc.), and system architecture (×86, arm64, etc.). The quantitative values represent a feature vector of hardware characteristics such as number of cores, clock speed, and dedicated RAM.

This case retrieving task analyzes the categorical and the quantitative similarities between old and new cases (step 2), as shown in Figure 3. First, it calculates the categorical similarity using the Hamming distance. This operation returns the most similar results (the lower distance the more similar) as a list of tuples associated with a tuple identifier. Those with a distance value of lower than the threshold *cat_thr* are selected for the quantitative similarity measurement. From this list, the system takes the tuple identifiers and extracts the quantitative values of each DNN accelerator, represented as feature vectors. Then, the cosine similarities between each feature vector and the new case feature vector are calculated. From these, if the highest value is higher than the threshold *quant_thr*, the solution is found (case reuse), otherwise the solution requires an adaptation (case adaptation). The knowledge database is initialized with empty cases and predefined resources, as explained in the next sections (DNN models, DNN IEs, set of rules for case adaptation). In this case, the new case goes to the case adaptation step, automatically.

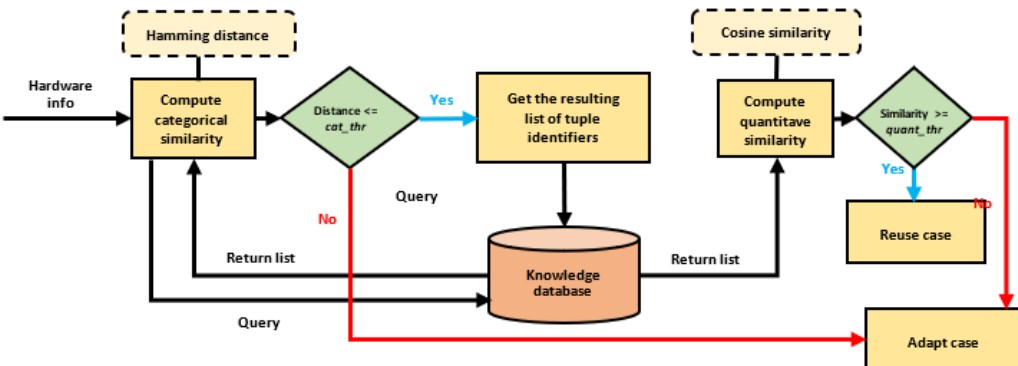

**Figure 3.** Case retrieving workflow.

### 3.2.2. Case Reuse/Adaptation

As was mentioned in the previous section, this task finds the solution of a problem by providing two alternatives, the reuse of a retrieved case solution or the adaptation of the new case solution (step 3). The case reuse only associates the previously retrieved solution to the new case problem. The case adaptation instead involves a more complex task. However, for both cases, the VPC creates a unified package of candidate DNN resources (DNN IEs, DNN models, and also a small dataset of labeled images for testing) and transfers it to the target device, robot, or gateway (step 4).

For case adaptation, we choose a rule-based engine to find the solution to the new case. We define a set of rules which represents the relationship between DNN IEs and DNN models. These rules are defined by an expert human (Machine Learning engineer). These defined rules and the hardware information of the new case are executed in the rule-based engine (expert system) and as a result, the expert system returns a list of possible candidates of DNN IE and DNN models, along with the testing dataset.

These DNN models may be of the same kind but are trained in different DNN IEs and with varying precision (from most accurate to fastest: FP32, FP16, or INT8). Some DNN IEs require the use of specific computing processors (e.g., Google's TFLite Edge TPU for Google's TPUs, or NVIDIA's TensorRT for NVIDIA's GPUs), but in other cases, the same processor is suitable for different DNN IEs. Besides, the FR solution is composed of several DNNs such as face and face-landmark detection, head pose, facial gesture, facial image quality analyzer, anti-spoofing detector, and identity recognition.

These models can be deployed into heterogeneous hardware. For example, some models could run in one CPU processor and the remaining models in the GPU processor, depending on the requests of image batches that suit the processor best. Here, with batch processing, we do not refer to resizing the input of the DNN model to a specific image batch size, but to a batch of several inference requests, which are executed asynchronously. Depending on the target hardware architecture, optimizing the inference processes, such as splitting the *N* batch of images in *N* requests, or using inference asynchronously could improve the performance. That is why selecting the optimal model configuration plays an essential role in the deployment process.

### 3.2.3. Case Evaluation

In step 5, the CBR system evaluates the solution proposed in the previous case reuse/adaptation task. More specifically, it evaluates the performance of the solution candidates of DNN IE and models. This task is done directly on the IoT device.

This evaluation process, apart from benchmarking the inference latency and accuracy of the DNN models, also analyzes the DNN graph partitioning capabilities for those models in the DNN IEs that have heterogeneous inference capabilities, such as OpenVINO and TensorRT [26]. The DNN graph partitioning consists of splitting the DNN model into

different subgraphs (group of layers) and delegating their workload to the most suitable DNN accelerators. The case evaluation algorithm is shown in Algorithm 1.

---

**Algorithm 1.** Case evaluation.

---

**Input:** Heterogenous hardware device configurations (**HCONF**),
Image/request inference batch sizes list (**IB**), DNN model candidates (**DMC**),
DNN IE (**DIE**), Testing dataset for benchmarking (**TEST_DATA**)
**Output:** List of optimal heterogeneous Hardware configuration per batch (**OHC**)

| | |
|---|---|
| 1 | For **batch** in **IB**: |
| 2 | For **hetero_device_conf** in **HCONF** |
| 3 | For **IE** in **DIE**: |
| 4 | **DM** = get_suitable_precision_DNN_models(**IE, DMC, hetero_device_conf**) |
| 5 | **DB** = load_database_for_benchmarking(**TEST_DATA**) |
| 6 | load_models_to_IE(**IE, DM, DB, hetero_device_conf**) |
| 7 | **Aff** = Get_estimated_layer_affinities_from_DM (**IE, DM, hetero_device_conf**) |
| 8 | If (all_layers_supported(**IE, hetero_device_conf, DM**) = OK) |
| 9 | make_benchmark(**IE, DM, hetero_device_conf, Aff, DB**) |
| 10 | *Perf_list_device* = Store_performance_metrics(**DM, hetero_device_conf, IB**) |
| 11 | **OHC**.append(**find_optimal_hconf_per_batch**(*Perf_list_device*, **IB**)) |
| 12 | Else |
| 13 | discard_device_configuration(DM, hetero_device_conf) |
| 14 | Return **OHC** |

---

This algorithm returns a list of optimal hardware configurations (OHC) for each batch size. The algorithm iterates through all possible batch sizes and, for each one, iterates through their corresponding heterogeneous hardware configuration (HCONF). For each pair of batch and downloaded configuration from the VPC, the algorithm loads the corresponding IE based on the hardware heterogeneous configuration and selects the required precision model according to the VPC downloaded candidate. Then, the algorithm chooses a suitable DNN model for benchmarking and the image testing database included in the VPC downloaded package. When the selected model is loaded, the algorithm checks if all the DNN layers are supported by the device. If all layers of the DNN model are suitable for the current DNN IE, the benchmark is performed, otherwise, the configuration is discarded. When the DNN graph partitioning is not supported by the device, this step is skipped, but the benchmarking is executed anyway for the evaluation process. Finally, for each batch size, the *find_optimal_hconf_per_batch* function selects the best DNN model configuration finding the trade-off between inference latency and accuracy results following the MLPerf benchmarking standard [6]. Then, this configuration is added to the OHC list.

### 3.2.4. Case Retaining

Steps 6 and 7 are undertaken to submit the selected configuration back to the VPC and then, this new case (problem and solution) is stored in the knowledge database—the case problem (categorical and quantitative values) and the case solution (the most optimal DNN IE and DNN model configuration).

### 3.2.5. Updating the New Trained DNN Models

Since the DNN models and DNN IEs are constantly evolving, we consider the updating process of these resources (step 8). This process does not only include uploading DNN resources, but also the rules and relations for the rule-based engine. Thus, following these new rules, DNN resources are stored in the knowledge database. This updating process also requires evaluating whether these new DNN resources are better suited for the IoT device. Thus, when the updating process is finished, the VPC notifies all IoT devices that new DNN resources are available.

A Machine Learning (ML) engineer should assume the responsibility of the updating process. The challenges for the ML engineer are as follows: to keep track not only of the characteristics of the deployed FR solution (DNN models and IEs) and the compatible hardware architecture, but also of the advances in the DNN-based technology that could be applied to improve the accuracy, robustness and/or performance. Besides, the ML engineer should update the CBR rules accordingly.

### 3.2.6. Running the Face Recognition System

When the system is ready to load the FR system, our approach provides the functionality to load the most optimal model. However, when the DNN model is loaded to the DNN accelerator memory, its graph cannot change during runtime. The model's performance depends on several factors such as its case scenario, input data throughput (number of face images to be processed simultaneously) and batch size, which leads to situations where models with larger batch sizes may not obtain the best results. In cases for which several faces need to be verified, configurations with larger batch sizes perform faster than other options with the highest precision models, and the face recognition system's workload will use the 100% of the hardware resources, which is the most optimal way to use them. Conversely, when the input face images throughput is low, choosing this option could lead to energy waste, degrading their performance. Furthermore, choosing cheaper hardware accelerators and reducing batch size can perform similarly to lower power consumption.

### 3.3. Biometric Data Management

The biometric data of the user results from processing the users' facial images with a DNN trained with data that correspond to other people, but not to the user. Thus, this biometric data is an abstract representation generated by a combination of the appearances of some of the people used for training the DNN, but not specifically from the user. This means that it would be impossible to reconstruct the user's real appearance with high precision from the biometric data, using techniques such as those described in [27]. Nevertheless, this kind of reconstruction could reveal relevant personal information from the real user, due to their similarity to those people, such as "gender", "skin color", "hairstyle", etc., which needs to be protected.

While a completely secure system against any kind of threats does not exist, an appropriate level of security for the targeted scenario and the expected kind of attacks can be designed. In our approach, biometric data would never be transferred to the cloud, i.e., it would only be stored in those devices where the FR solution has been deployed. Thus, an expected possible attack could be the access of somebody to the devices, robots, or gateways to steal the stored data. To prevent this, the data should be encrypted, and the encryption keys should be kept safe by an administrator.

In our context, we consider the method in [28] as the most appropriate for our purpose. It proposes an efficient fully homomorphic encryption-based approach to cryptographically secure the registered and the probe biometric data, performing the matching directly in the encrypted domain, leveraging the observation that a typical face matching metric, either Euclidean distance or cosine similarity, can be decomposed into its constituent series of addition and multiplication operations. It utilizes a batching scheme that allows for the homomorphic multiplication of multiple values at the cost of a single homomorphic multiplication, and dimensionality reduction to further provide a trade-off between computational efficiency and matching performance. Therefore, the biometric data shared among the devices, robots, and gateway remain encrypted, so long as the administrator keeps the encryption keys safe (e.g., in a secure hardware element [29]).

This approach requires four keys, namely, a public key for encryption, a private key for decryption of the scores, and linearization and Galois keys for the matching operations of the encrypted data (Figure 4). In this way, if someone steals the stored biometric data from the hardware platform, it would not be usable without access to the private key required for decryption. This private key should be kept safely in a secure element of

the hardware platform, for example, in a Trusted Platform Module (TPM) or a Trusted Execution Environment (TEE). TPMs are normally available in modern computer PC motherboards. TEEs are available in Jetson Xavier NX, Jetson AGX Xavier series, and Jetson TX2 series devices.

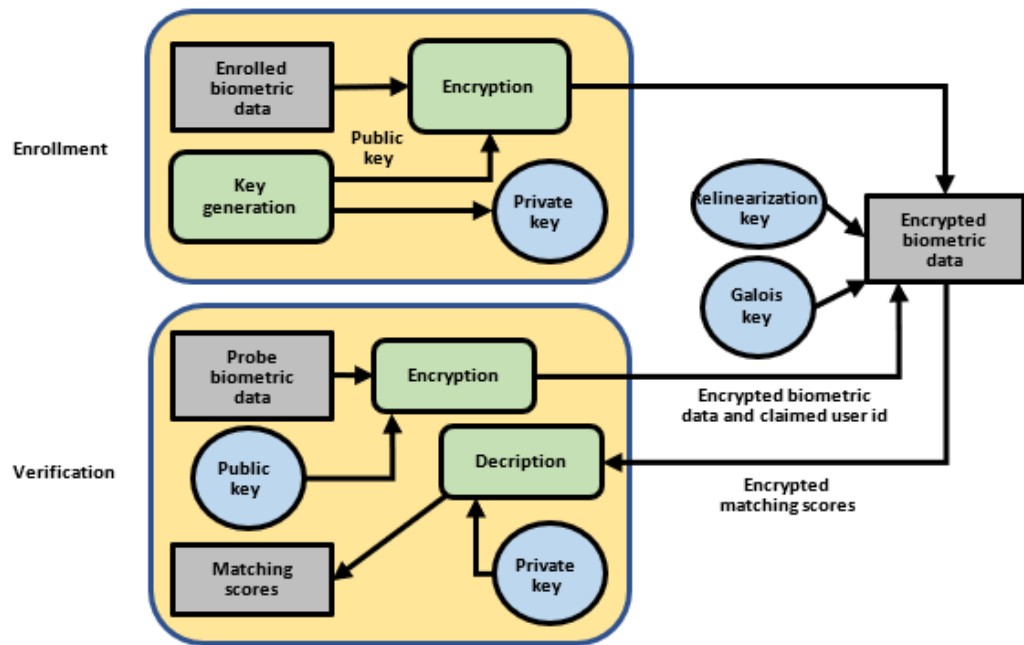

**Figure 4.** Biometric data management with fully homomorphic encryption during enrollment and verification.

## 4. Results and Discussion

### 4.1. Qualitative Evaluation

Table 1 shows a qualitative comparison of our approach with respect to state-of-the-art alternatives, in terms of face recognition technology, interaction assistance, deployment algorithms and privacy. We also include, in this comparison, our previous study [7] to illustrate, in more detail, the differences. As can be observed, both [7] and the improved approach presented in this paper stand out in terms of interaction assistance and deployment, issues which were poorly addressed by the other alternatives. The main differences between [7] and the extended version are the inclusion of a DNN for image quality assessment, which allows for an improved interaction with the user, and especially, the way in which the DNN resources are deployed, by means of the presented knowledge-driven approach, which takes advantage of previous cases (experience) to reuse and adapt the knowledge to new cases, as required. The versatility offered by our deployment scheme improves on previous approaches by removing dependence on the devices' hardware and allowing for the usage of more complex DNN models with improved FR capabilities. In terms of privacy and data protection, our method employs an efficient and fully homomorphic encryption-based approach and restricts all operations to local devices, offering a lightweight processing pipeline without compromising security. All these features make our approach well-suited to all demands required by an industrial scenario to be deployed among workers.

**Table 1.** Comparison of state-of-the-art IoT platform approaches vs. our proposal (FD: Face detection, FLD: Face and facial landmark detection, PGR: Pose and gesture recognition, IQA: image quality analysis, SAD: spoofing attack detection, FIR: Facial identity recognition).

| Method | Face Recognition Approach | Assistance for Interaction | Deployment of Algorithms | Privacy in the IoT Platform |
|---|---|---|---|---|
| [15,16] | Pretrained Haar-based model for FD and LBP features for FIR to be trained on the cloud. | Not considered | Manually predefined | Schemes for authentication, session key agreement, data encryption, and data integrity checking for secure data transmission and storage. |
| [17] | Discriminative dictionary learning for FIR that needs to be trained. | Not considered | Manually predefined | Biometric data encrypted with a low complexity encrypting algorithm based on random unitary transformation. |
| [18] | DNN for FIR split in two parts: one deployed on the user side and the other on the edge server side. | Not considered | Manually predefined | Differential privacy for user's confidential datasets. No cryptographic tools used to keep user side lightweight. |
| [7] | Pretrained DNN models for FLD, PGR, SAD, and FIR deployed on fog gateway and client devices suitable for DNN inference. | Real-time visual feedback based on FLD and PGR to guide the user during enrollment and verification. | Automated selection of the appropriate DNN inference engine, DNN model configurations, and batch size, based on IoT device characteristics. | Biometric data homomorphically encrypted. All computations are performed on the private network. Biometric data not sent to the cloud. |
| Ours | Pretrained DNN models for FLD, PGR, IQA, SAD, and FIR deployed on fog gateway and client devices suitable for DNN inference. | Real-time visual feedback based on FLD, PGR and IQA to guide the user during enrollment and verification. | Automated selection of the appropriate DNN inference engine, DNN model configurations, and batch size, by means of a knowledge-driven approach. | Biometric data homomorphically encrypted. All computations are performed on the private network. Biometric data not sent to the cloud. |

## 4.2. Quantitative Evaluation

In order to evaluate the performance achieved by a DNN-based FR solution following our deployment approach, compared to standard—manually predefined—deployments, we performed two experiments on two different target devices. We used MobileNetV1 [30], ResNet-50 [31], and SSD-MobileNetV1 [32], which are popular DNN architectures for the computer vision tasks of "image classification" (similar to all the tasks applied to cropped facial images; spoofing detection, identity recognition, etc.), and "object detection" (face regions in our case). The weights of these DNNs have FP16 precision. We used the following hardware for the benchmarking experiments:

- IEI TANK AIoT Developer Kit embedded PC with Intel Mustang V100 MX8 for DNN acceleration card. This hardware contains an Intel CPU, GPU and a High Density Deep Learning (HDDL) card (Mustang) compatible with Intel's OpenVINO DNN IE.
- NVIDIA Jetson Xavier AGX 32GB. This hardware contains a NVIDIA GPU with 512-core NVIDIA Volta™ GPU with 64 Tensor cores and 2 NVDLA (dla0, dla1) DNN accelerators compatible with NVIDIA's TensorRT DNN IE.

The first experiment consists of measuring the performance of MobileNetV1 and ResNet-50 for different image batch sizes (1 to 48) with different inference configurations, assuming that all hardware resources are fully available. These two hardware platforms allow DNN graph partitioning across heterogeneous hardware, given a selection of computing processors and their execution priority. For the TANK, we considered the following configurations: <CPU, HDDL>, <GPU, CPU> and <HDDL, GPU, CPU>, where the order represents the execution priority. For the Jetson, we selected the following configurations: <GPU>, <DLA0, GPU>, <DLA1, GPU>. The current version of Jetpack (4.5.1)—the Jetson's SDK—does not allow for the deployment of a DNN model across both DLA cores (DLA0 and DLA1). Each DLA core is totally independent and can only communicate with the GPU for inference.

Figure 5 shows that the <GPU, CPU> configuration performs best with smaller batches (1 to 4), while <HDDL, GPU, CPU> is the fastest for higher batch sizes. Those would be the configurations chosen by our approach for the respective batch sizes (the optimal). In principle, the CPU's and GPU's hardware processing capabilities are bigger than those of the Mustang V100. Thus, theoretically, the <GPU, CPU> configuration should be the best choice for all batch size configurations. However, the parallelization of the processing methods of the CPU, based on SIMD vectorization techniques, are designed for a general parallelization purpose, and not for DNN inference processing. Additionally, the cache L1/L2/L3 memory is shared across the CPU and GPU hardware, so, this creates a time overhead because of the data transference between the cache memory to CPU and GPU. In contrast, the Mustang V100 MX8's architecture is specifically designed for DNN inference processing, with 8 cores of Myriad X processors ensembled at USB 3.2 speed. The processing capabilities of each core are much lower than those of the CPU and GPU, but when several inference instances are executed, it runs more efficiently. Besides, each Myriad X core has a dedicated memory (512MB for each core) to load DNN model weights. Thus, the inference processing is parallelizable at 100% and there is not data transfer overhead.

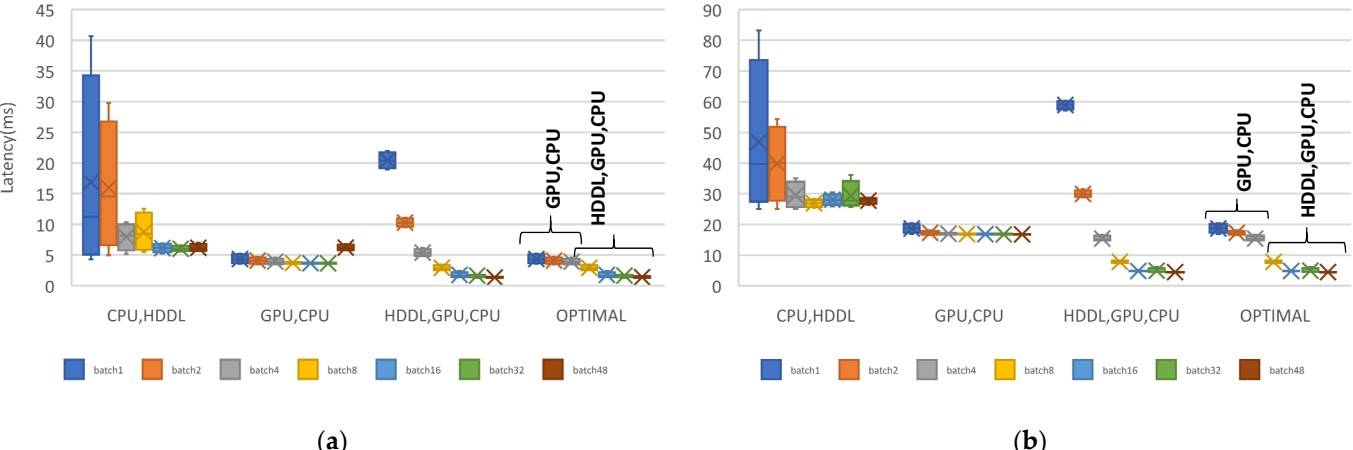

(**a**)                (**b**)

**Figure 5.** Comparison between DNN inference performances obtained by the heterogeneous deployment optimizer vs. manual heterogeneous configurations on a TANK AIoT Dev. Kit with a Mustang-V100-MX8 DNN acceleration card.: (**a**) MobileNetV1; (**b**) ResNet-50.

Figure 6 shows that, in the Jetson, more stable inference results are obtained, with a smaller standard deviation of latency times for all batches and configurations. It also shows that the GPU performs better than the <DLA, GPU> configuration. One of the main reasons is because the capacity of each DLA is 2.5 TOPS, while the GPU can execute 11 TOPS with FP16 precision. Additionally, even though the DLA's design is based on four highly configurable modules (convolution, normalization, activations, data transfer), deploying DNNs to this hardware faces the following drawbacks: first, the current implementation of this module has a limited number of DNN layers and thus those not supported are transferred to the GPU adding an important overhead. Moreover, each DLA core is only able to execute four batch streams in parallel. Therefore, even though this NVDLA architecture is innovative, and its low power consumption (0.5–1 Watt) is interesting for IoT platforms, it is still far from the GPU's processing capabilities. In contrast, the NVIDIA Volta design contains eight Streaming Multiprocessors (SM) and each SM includes 64 CUDA cores and eight Tensor Cores. Additionally, it has ultra-fast memory access with 128 KB of L1 cache memory per Volta SM and sharing 512 KB L2 offering faster access than previous generations.

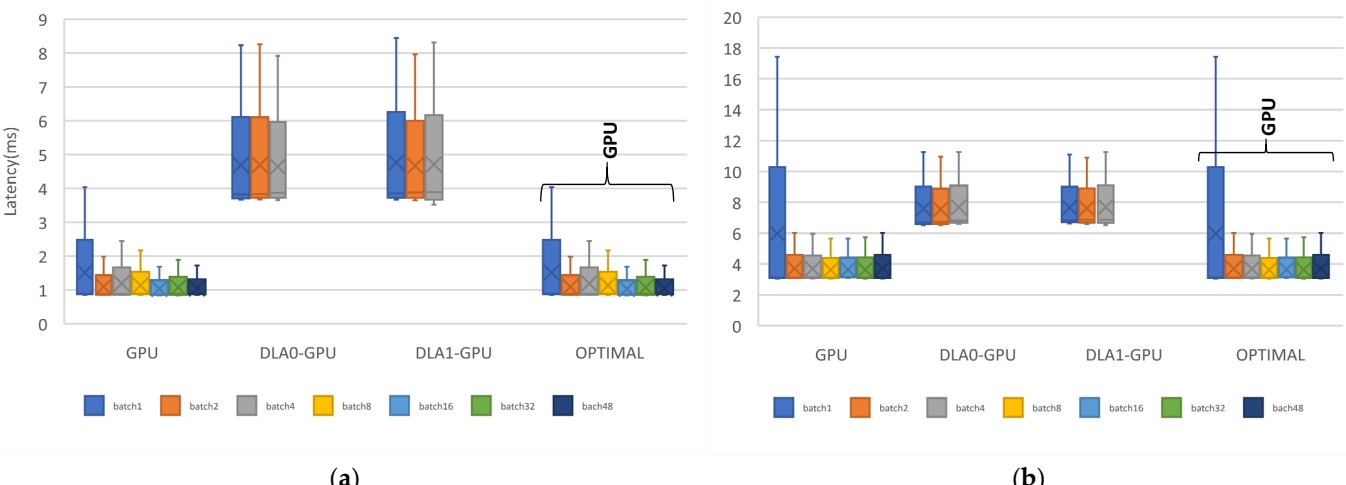

(**a**)　　　　　　　　　　　　　　　　　　(**b**)

**Figure 6.** Comparison between DNN inference performances obtained by the heterogeneous deployment optimizer vs. manual heterogeneous configurations on Jetson Xavier AGX 32GB.; (**a**) MobileNetV1; (**b**) ResNet-50.

The second experiment consisted of measuring the performance of MobileNetV1 and ResNet-50 for different image batch sizes with different inference configurations, but with SSD-MobileNetV1 constantly executed as background "computing noise" in different computing processors. Figures 7 and 8 show the influence of this noise and the configurations that would select our approach in each case. This influence of the background execution is clearly visible in Figure 7. For example, when the CPU is selected for background execution, the <GPU, CPU> and <HDDL, GPU, CPU> devices are selected as the optimal configurations. However, when the GPU is selected for background execution, the HDDL heterogeneous configuration takes almost all batch configurations as optimal (4, 8, 16, 32, 48).

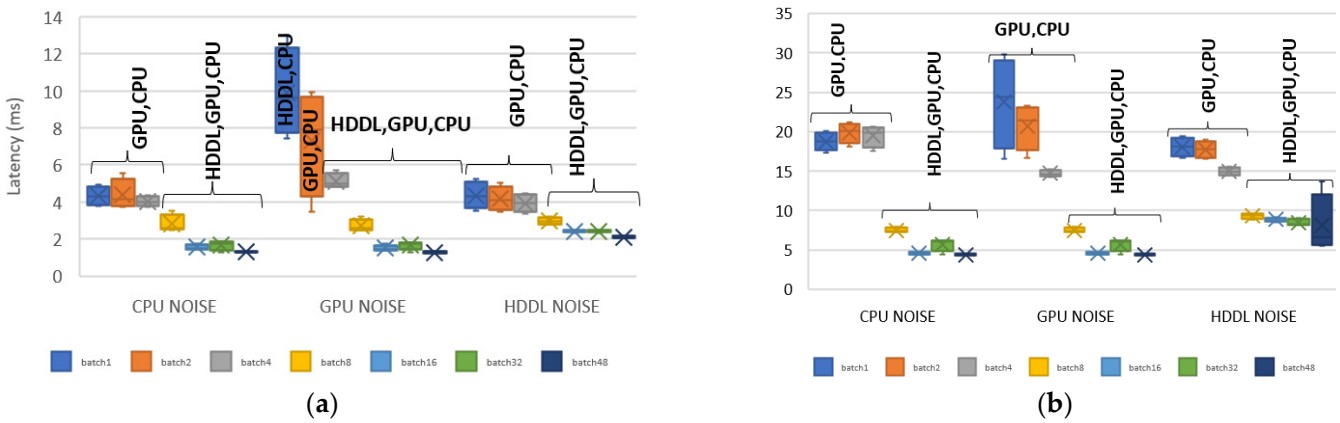

(**a**)　　　　　　　　　　　　　　　　　　(**b**)

**Figure 7.** The influence of the background hardware usage with heterogeneous deployment optimizer decisions on TANK AIoT Dev. Kit with a Mustang-V100-MX8 DNN acceleration card: (**a**) MobileNetV1; (**b**) ResNet-50.

Figure 8 instead, shows very stable results such as those shown in Figure 6, and the GPU configuration is always the optimal choice. However, as was expected, the higher latency times on both DNN models are clearly visible because of the background execution of SSD-MobileNetV1.

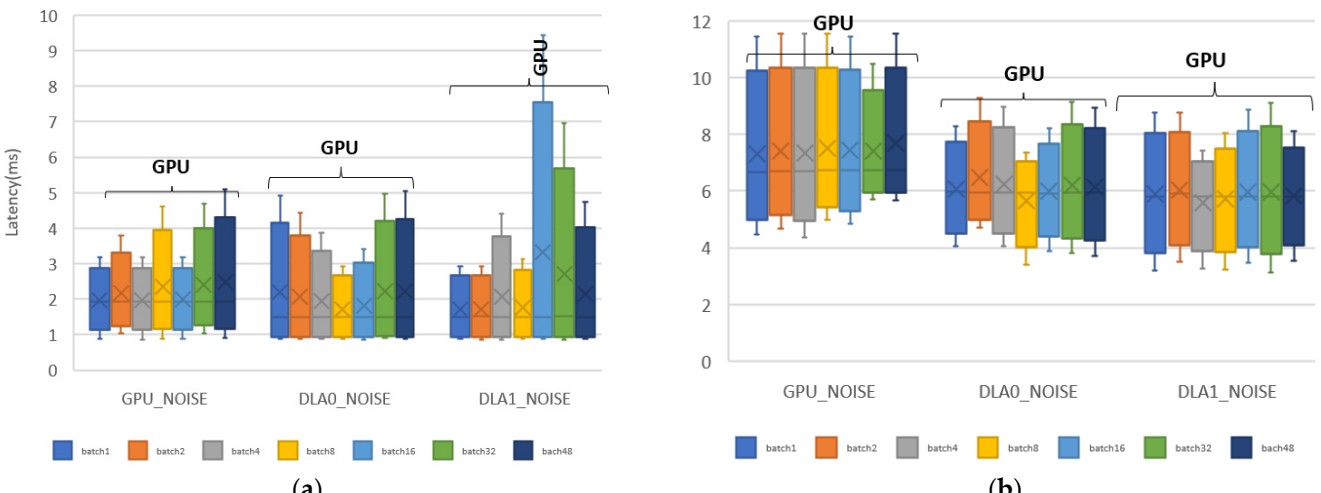

**Figure 8.** The influence of the background hardware usage with heterogeneous deployment optimizer decisions on TANK AIoT Dev. Kit with a NVIDIA Jetson Xavier AGX: (**a**) MobileNetV1; (**b**) ResNet-50.

The AHDO addresses the challenges of automatically selecting the most optimal DNN models and IEs across the wide variety of heterogeneous hardware devices. The DNN-based technology (i.e., models, IEs, and hardware architectures) is constantly evolving and the selection of the most suitable DNN package for each hardware architecture is becoming more complex. We manage this automatic decision-making process by creating a CBR system to define the rules and case structure. Defining the rules for a CBR system includes the challenge of understanding the problem and requires the integration of a reasoning engine into the AHDO. In addition, we have designed an algorithm for the case evaluation task to analyze the performance (latency, accuracy, layer partitioning) and to ensure that the selected DNN package is the most adequate. The presented results reveal how this AHDO algorithm improves the DNN package selection process. This automatic deployment of DNN packages is generalizable beyond the FR-related DNN models. Moreover, our approach allows for the deployment of DNN-based FR solutions in heterogeneous IoT platforms that could be used for different kinds of applications in Industry 4.0, the medical domain, etc. In the end, each application has its specific requirements for the FR solution (e.g., real-time processing, low latency, small face detection, high accuracy for landmark detection, etc.), which need to be analyzed by the ML engineer to prepare the most appropriate FR pipeline and DNN package in each case.

## 5. Conclusions

In this work, we presented a knowledge-driven approach for the optimal deployment of DNN-based FR solutions in a heterogeneous IoT platform for industrial applications. Our approach tackles its specific challenges in terms of ease of use, hardware heterogeneity, and security. Ease of use is covered by a customized workflow that offers an intuitive interface that considers the shared use between the user and companions, by means of automated feedback to automatically take the "selfie" with the best possible quality. Device heterogeneity is addressed by a smart deployment optimizer, capable of selecting the appropriate DNN inference approach for the targeted device's hardware specifications. Data security is enforced by a scheme that avoids the transmission and the storage of biometric data in the cloud and employs a fully homomorphic encryption to perform all face-matching operations directly on the encrypted domain. A qualitative comparison with state-of-the-art alternatives and preliminary experiments to select the optimal inference approach for different computing situations (different batch sizes and computing noise) revealed its potential for the predefined objective. Future work will extend the performance evaluation of the AHDO deployment process with new DNN model architectures, IEs, and IoT devices (mobile phones, embedded devices, FPGAs). Additionally, we will analyze

other knowledge-driven approaches for automatic deployment, and conduct a more in-depth study for user interaction issues.

**Author Contributions:** Conceptualization, U.E. and L.U.; methodology, U.E., L.U. and C.L.; software U.E., L.U. and J.G.; validation, U.E., L.U. and C.L.; formal analysis, U.E.; investigation, U.E., L.U. and C.L.; writing—original draft preparation, U.E., L.U. and C.L.; writing—review and editing, U.E., L.U., C.L., J.G., N.A., A.B. and I.A.-C.; supervision, L.U. and I.A.-C.; project administration, L.U.; funding acquisition, L.U. All authors have read and agreed to the published version of the manuscript.

**Funding:** This work was supported by the SHAPES project, which has received funding from the European Union's Horizon 2020 research and innovation program under grant agreement no. 857159, and in part by the Spanish Centre for the Development of Industrial Technology (CDTI) through the Project ÉGIDA—RED DE EXCELENCIA EN TECNOLOGIAS DE SEGURIDAD Y PRIVACIDAD under Grant CER20191012.

**Institutional Review Board Statement:** Not applicable.

**Informed Consent Statement:** Not applicable.

**Data Availability Statement:** Not applicable, the study does not report any data.

**Conflicts of Interest:** The authors declare no conflict of interest.

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
