# Peer review of "Designing Automated Deployment Strategies of Face Recognition Solutions in Heterogeneous IoT Platforms"

_information, doi:10.3390/info12120532_

Round 1

Reviewer 1 Report

The paper presents a comprehensive methodology for the automated deployment of DNN-based FR solutions in IoT devices. The authors address an important problem and I think the paper is relevant and interesting for a wide readership. The article is well and clearly written. The concept presented makes sense and gives a good summary of the challenges to be overcome. The results presented, however, remain preliminary. A couple of open questions remain regarding the actual implementation of the methodology (e.g. what could be practical challenges regarding the AHDO unit? Challenges for the ML engineer? Initialisation of the knowledge database? Etc.).  The authors could address these issues in more detail in the concluding discussion and outline a more detailed plan for future research.  

In summary, I believe that the article is valuable contribution to the field and I support its publication.

Reviewer 2 Report

The contribution presents the design of automated deployment strategies of face recognition solutions in heterogeneous Internet of Things (IoT) platforms. In summary, the paper is well written and fits within the scope of Information. Please add a discussion if your approach can also be used in the medical domain, like in this work, where a face detection has been used for a subsequent Augmented Reality (AR) application in facial surgery:
https://link.springer.com/article/10.1007/s10278-019-00272-6
